# Comparative Analysis of Herbaceous and Woody Cell Wall Digestibility by Pathogenic Fungi

**DOI:** 10.3390/molecules26237220

**Published:** 2021-11-28

**Authors:** Yanhua Dou, Yan Yang, Nitesh Kumar Mund, Yanping Wei, Yisong Liu, Linfang Wei, Yifan Wang, Panpan Du, Yunheng Zhou, Johannes Liesche, Lili Huang, Hao Fang, Chen Zhao, Jisheng Li, Yahong Wei, Shaolin Chen

**Affiliations:** 1College of Life Sciences, Northwest A&F University, Yangling, Xianyang 712100, China; yhdou_iris@nwafu.edu.cn (Y.D.); mund.nitesh@gmail.com (N.K.M.); weiyanping@genecompany.com (Y.W.); yisliu2021@126.com (Y.L.); weilf@nwafu.edu.cn (L.W.); wyfyffs@126.com (Y.W.); tkj@nwafu.edu.cn (P.D.); zhouyunheng@nwafu.edu.cn (Y.Z.); liesche@nwafu.edu.cn (J.L.); fanghao@nwafu.edu.cn (H.F.); chenzhao@nwafu.edu.cn (C.Z.); lijsh2011@163.com (J.L.); 2Biomass Energy Center for Arid and Semi-Arid Lands, Northwest A&F University, Yangling, Xianyang 712100, China; 3College of Chemistry and Chemical Engineering, Shanxi Datong University, Datong 037009, China; DTDXHXSHD@126.com; 4College of Plant Protection, Northwest A&F University, Yangling, Xianyang 712100, China; lilyhuangk@163.com; 5Shaanxi Key Laboratory of Agricultural and Environmental Microbiology, Northwest A&F University, Yangling, Xianyang 712100, China

**Keywords:** cellulose, hemicelluloses, lignin, apple tree branch, wheat straw, rapeseed straw, digestibility, pathogenic fungi, CAZyme, cellulase, hemicellulases

## Abstract

Fungal pathogens have evolved combinations of plant cell-wall-degrading enzymes (PCWDEs) to deconstruct host plant cell walls (PCWs). An understanding of this process is hoped to create a basis for improving plant biomass conversion efficiency into sustainable biofuels and bioproducts. Here, an approach integrating enzyme activity assay, biomass pretreatment, field emission scanning electron microscopy (FESEM), and genomic analysis of PCWDEs were applied to examine digestibility or degradability of selected woody and herbaceous biomass by pathogenic fungi. Preferred hydrolysis of apple tree branch, rapeseed straw, or wheat straw were observed by the apple-tree-specific pathogen *Valsa mali*, the rapeseed pathogen *Sclerotinia sclerotiorum*, and the wheat pathogen *Rhizoctonia cerealis*, respectively. Delignification by peracetic acid (PAA) pretreatment increased PCW digestibility, and the increase was generally more profound with non-host than host PCW substrates. Hemicellulase pretreatment slightly reduced or had no effect on hemicellulose content in the PCW substrates tested; however, the pretreatment significantly changed hydrolytic preferences of the selected pathogens, indicating a role of hemicellulose branching in PCW digestibility. Cellulose organization appears to also impact digestibility of host PCWs, as reflected by differences in cellulose microfibril organization in woody and herbaceous PCWs and variation in cellulose-binding domain organization in cellulases of pathogenic fungi, which is known to influence enzyme access to cellulose. Taken together, this study highlighted the importance of chemical structure of both hemicelluloses and cellulose in host PCW digestibility by fungal pathogens.

## 1. Introduction

Production of biofuels and sustainable bioproducts from plant biomass benefits the environment by reducing greenhouse gas emissions [1]. However, low efficiency of enzymatic hydrolysis of cellulosic biomass to fermentable sugars is still a major challenge for cost-effective production of biofuels and bioproducts [2,3,4]. Further improvement of this process depends on a deeper understanding of the relationship between biomass cell wall structure and enzymatic digestibility [5,6].

Cell wall material from the plant species, such as monocot grasses, herbaceous dicots, and woody dicots, has been used as candidate biomass feedstocks for the production of biofuels and bioproducts [7]. Despite great progress in investigating PCW recalcitrance, few studies have compared the enzymatic digestibility of herbaceous and woody biomass by pathogenic fungi. Different woody and herbaceous biomass have distinguishable PCW compositions, polymer features, and structures [8,9]. For instance, substitutions on xylans, the major component of the secondary cell walls (SCWs) of dicotyledonous and grass plants, are diverse and vary phylogenetically [10,11]. Eudicot xylans are acetylated, whereas monocot xylans are acetylated and highly arabinosylated [12]. The different substitutions affect both the self-association properties of the xylan polymer and its interaction with cellulose and lignin [13,14].

To compare the impact of hemicelluloses and lignin on the digestibility of herbaceous and woody plant cell walls, we assessed the hydrolysis profile of plant biomass from herbaceous monocots (wheat and switchgrass straw), herbaceous dicot (rapeseed straw), and woody dicot (apple tree branch) by selected plant pathogens before and after hemicellulase or peracetic acid (PAA) pretreatment. Hemicellulase pretreatment makes changes to not only the main chains of polysaccharides but also branching decoration with sugars (i.e., mono- or oligosaccharides) or non-sugars (i.e., acetylation, methylation, or feruloylation) in main chains [15]. PAA, a relatively inexpensive reagent, can reduce the degree of polymerization of lignin and increase its solubility in water at relatively low temperatures [16]. After PAA pretreatment, which exposed surface fibrillar cellulose, field emission scanning electron microscopy (FESEM) was applied to examine the differences of surface cellulose organization between woody and herbaceous cell wall substrates. 

Effects of hemicellulase and PAA pretreatment on host substrate preferences were further compared with their pathogenic fungi (Table 1). In addition to the promising energy crop switchgrass (SG), wheat straw (WS) and rapeseed straw (RS) were used as representative residues of major monocot and dicot crops in Northwest China [17,18]. Wood branches were rich in certain areas and apple tree branch (AB) was selected as it has the largest production in non-forest areas in Northwest China [19]. *Valsa mali*, an apple-tree-specific pathogen [20], was used as a woody plant pathogen. *Botryosphaeria dothidea*, which infects various woody species including apple tree [21,22], was used as a comparison to the apple-tree-specific *V. mali*. Another comparison was made with *Colletotrichum gloeosporioides*, which infects a broad range of hosts, including woody and herbaceous plants [23]. *Sclerotinia sclerotiorum* was used as a representative herbaceous dicot pathogen that infects rapeseed [24], while *Bipolaris sorokiniana* [25,26], *Gaeumannomyces graminis* [27], *Fusarium graminearum* [28,29,30], and *Rhizoctonia cerealis* [31,32] were used as herbaceous monocot pathogens that infect wheat. Comparing the hydrolytic activities of the selected pathogens would help determine whether the pathogens have host preferences. Pretreatment by PAA and hemicellulases helps examine if lignin and hemicelluloses impact the digestibility of host PCWs by pathogenic fungi. Our previous studies suggest that mutations in cellulose synthases influence not only cellulose synthesis and deposition [33,34] but also PCW porosity [35], a critical parameter affecting the accessibility of cellulose to cellulolytic enzymes [36]. To examine if cellulose contributes to host PCW digestibility, we visualized cellulose organization in the PCW biomass tested using FESEM and analyzed the sequences and domain organization of GH6 and GH7 cellobiohydrolases (CBHs), the major components of fungal cellulases [37]. Taken together, this initial investigation aims to provide a basis for further exploration of the interactive relationships among cellulose, hemicellulose, and lignin in host PCWs and PCWDEs evolved by pathogenic fungi to overcome the recalcitrance of PCWs.

## 2. Results and Discussion

### 2.1. Extracts from Valsa mali, B. sorokiniana, and S. sclerotiorum Showed Significant Hydrolytic Preferences for Apple Tree Branch, Wheat Straw, and Rapeseed Straw, Respectively

To compare the digestibility of herbaceous and woody PCWs by pathogenic fungi, the biomass residues selected include apple tree branch (woody dicot), rapeseed straw (herbaceous dicot), and wheat and switchgrass straw (herbaceous monocot). *Valsa mali* was used as a representative woody pathogen as it specifically infects apple trees [20]. PCWDEs from *V. mali* and other pathogenic fungi were extracted from the cultures grown on switchgrass, which has been shown to have a similar PCWDE induction profile to other substrates [39]. As shown in Figure 1A, the extracts from *V. mali* showed a significant preference for the biomass of apple tree branch (AB), with approximately one-fold higher lignocellulolytic activity on AB than that on wheat straw (WS), switchgrass straw (SW), and rapeseed straw (RS) (Figure 2). A significant preference of *V. mali* for the biomass of AB correlated with the host preference of this pathogen for apple trees (Table 1) [20]. The extracts of *C. gloeosporioides* also showed greater hydrolysis in AB than WS, SW, and RS, while this pathogenic fungi infects not only apple trees and other woody plants, but also herbaceous plants [23]. By contrast, the extracts of *B. dothidea*, another common pathogen that occurs on a large number of hosts [22], did not show higher hydrolysis in AB but SW and RS. On the other hand, the extracts of pathogens for cereals and grasses, including *G. graminis*, *F. graminearum*, and *R. cerealis*, exhibited greater hydrolysis on the cereal and grass biomass tested (i.e., WS, SG, and RS) than the woody biomass of AB (Figure 1A). In particular, the extracts of *B. sorokiniana* and *S. sclerotiorum* showed significant host preferences for WS and RS, respectively. Taken together, the results corroborate the finding that hydrolytic preferences for biomass type were generally correlated with host preferences of the plant pathogens tested [39].

### 2.2. Hemicellulase Pretreatment Modulated Hydrolytic Preferences of Pathogenic Fungi for Biomass Type

As shown in previous studies [40,41], hemicellulases, such as xylanases and feruloyl esterases, are required for virulence of some plant pathogens. Disruption or silencing of xylanase genes in *V. mali* and other plant pathogens have been shown to cause defects in virulence [40,42,43]. Feruloyl esterases, a subclass of carboxylic esterases (CEs), cleave ester bonds that crosslink hemicellulose and lignin. Deletion of feruloyl esterase genes in *V. mali* and other plant pathogens also caused a reduction in pathogenesis [41,44].

To examine if hemicellulases play a role in hydrolytic preferences of the pathogenic fungi tested, biomass substrates were pretreated by hemicellulases from the saprophytic fungus *Aspergillus niger* (Hemicellulase H2125, Sigma), which contains a mixture of xylanases and other hemicellulases. Compared to no pretreatment, hemicellulase pretreatment caused significant changes in hydrolytic preferences of the pathogens tested. For instance, hemicellulase pretreatment switched the hydrolytic preference of *C. gloeosporioides* from AB to WS, while the pretreatment switched the preference of *G. graminis* from WS, SG, and RS to WS (Figure 1B). It was also notewothy that hemicellulase pretreatment showed differential effects on the digestibility of PCW substrates (Figure 3A,B). The pretreatment reduced the hydrolytic activities of *F. graminearum* on all the crop residues tested. The antagonistic effect was also observed with *B. dothidea* on SG and AB, *B. sorokiniana* on SG, *C. gloeosporioides* on WS and AB, *G. graminis* on SG and RS, *R. cerealis* on SG and AB, *S. scierotiorum* on SG and AB, and *V. mali* on AB (Figure 3B). On the other hand, synergistic effects of the pretreatment on PCW digestibility were observed with *B. dothidea* on WS and AB, *B. sorokiniana* on WS, RS and AB, *C. gloeosporioides* on RS, *R. cerealis* on WS, and S. scierotiorum on WS. The results indicate diversity of hemicelluloses among herbaceous and woody plant SCWs, corroborating the idea that the molecular heterogeneity of hemicelluloses plays a key role in modulating the interactions with cellulose microfibrils [45] and the occurrence of covalent linkages with lignin [12].

The composition of the crop residues was also determined (Figure 4), and the derived contents of cellulose, hemicelluloses, and lignin were in the range reported in previous studies [46,47,48,49,50,51,52,53,54]. While slightly reduced levels of hemicelluloses were observed in the herbaceous residues of WS (Figure 4A), SG (Figure 4B), and RS (Figure 4C), hemicellulase pretreatment had no significant effect on the level of hemicelluloses in the woody residue of AB (Figure 4D). The observed differential effects of hemicellase pretreatment on the digestibility of the woody residue AB by the pathogenic fungi tested cannot simply be explained by changes in hemicellulose composition. Hemicellulases are known to cleave or modify both main chains and side chains of hemicelluloses [11], which may lead to changes in interactions or crosslinking of hemicelluloses with cellulose microfibrils [45] and lignin [12], and thus modulated porosity of PCWs or accessibility of cellulolytic enzymes to cellulose microfibrils [5].

### 2.3. Delignification by PAA Pretreatment Increased Enzymatic Hydrolysis of Crop Residues and the Increase Was Generally More Profound with Non-Host than Host Plant Biomass

To examine the effect of delignification treatment on hydrolytic preferences of pathogenic fungi for crop residues, peracetic acid (PAA) was used to treat the biomass residues tested, which can reduce the degree of polymerization of lignin and increase lignin solubility in water [16]. Compared to no pretreatment (Figure 1A), PAA pretreatment also changed hydrolytic preferences of pathogenic fungi for the crop residues tested (Figure 1C and Figure 3D), but differently from hemicellulase pretreatment (Figure 1B and Figure 3B). While PAA pretreatment increased the digestibility of both the woody residue AB and the herbaceous residues WS, SG, and RS, differential effects of PAA pretreatment on host and non-host plant biomass were observed, as reflected by a more significant increase in the digestibility of non-host plant substrates than those of host plant substrates (Figure 3D). For instance, by the herbaceous pathogens, such as *B. sorokiniana, F. graminearum*, *G. graminis*, and *R. cerealis*, PAA pretreatment had a more significant effect on increasing the enzymatic hydrolysis of the woody substrate AB than that of the herbaceous substrates WS, SG, and RS. On the hand hand, by the woody pathogen *V. mali*, PAA pretreatment showed a more significant enhancement effect on the enzymatic hydrolysis of the herbaceous substrate RS than that of the woody substrate AB. Similarly, PAA pretreatment had a more significant effect on increasing the hydrolysis of non-host substrates such as WS, SG, and AB than those of the host substrate RS by *S. sclerotiorum*. Taken together, the differential effects of PAA pretreatment on host and non-host plant substrates reflect a key role of lignin in the diversity of cell walls among different plant species, which corroborates the idea that interactions and crosslinking among lignin, hemicelluloses, and cellulose affect emergent properties of PCWs such as mechanical strength, flexibility, and recalcitrance to enzymatic digestion [5,12,55].

### 2.4. Cellulose Organization May Contribute to Enzymatic Digestibility of Plant Cell Walls by Pathogenic Fungi

Our previous study using a fluorescence quenching assay revealed that mutations in cellulose synthases (CESAs) affect the nano-scale porosity of plant cell walls [35], a critical parameter determining PCW digestibility or degradability [36]. In this study, FESEM imaging was applied to examine if cellulose organization varies in different plant species. PAA treatment was applied to remove surface lignin to expose surface fibrillar cellulose. FESEM images revealed the differences of cell wall surface morphology between the woody AB and the herbaceous WS and RS. The surface pattern of cellulose fibrils appeared to be a relatively interwoven arrangement in the herbaceous WS and RS, while a relatively compact and parallel arrangement of cellulose fibrils was observed in the woody AB. Splitting of cellulose firbils caused by biomass grinding was observed in AB cell walls (Figure 4E), confirming the compact and parallel arrangement of cellulose fibrils in the SCW of woody species [56]. All together, the results suggest that chemical structure of cellulose may also contribute to enzymatic digestibility of PCWs by pathogenic fungi. This idea corroborates with studies showing that sequences of cellulose synthase (CESAs) domains, involved in the formation of CESA complexes [57], vary in plant species [58] and that phosphorylation of CESAs in these domains modulates cellulose synthesis and deposition [33,34].

Variation in chemical structure of cellulose in host plants is also reflected by domain organization and sequences of cellulases of pathogenic fungi. Representative cellulases in cellulolytic fungi include CBH1 and CBH2 [59]; their relative abundance is 60% and 20% in the exoproteome of *T. reesei*, respectively [37]. CBH1 and CBH2 belong to GH7 and GH6 families. Cellulose binding domain (CBD) (carbohydrate binding module 1) in GH6 or GH7 of the pathogenic fungi tested is generally at the N- or C-termini, respectively (Appendix A). According to CBD domain organization, GH6 and GH7 can be categorized into six groups (Figure 5A): (i) all GH6 and GH7 associated with a CBD (e.g., *T. reesei*); (ii) all GH6 and some GH7 with a CBD but other GH7 without a CBD (e.g., *F. graminearum*, *S. sclerotiorum*); (iii) some GH6 and GH7 with a CBD but others without a CBD (e.g., *G. graminis*, *N. crassa*); (iv) all GH6 with a CBD but GH7 without a CBD (e.g., *B. dothidea*, *V. mali*); (v) some GH6 with a CBD but other GH6 and all GH7 without a CBD (e.g., *C. gloeosporioides*); (vi) all GH6 and GH7 without a CBD (e.g., *B. sorokiniana*). It is noteworthy that the tree pathogens tested, including *V. mali* and *B. dothidea*, were categorized into group IV. 

In addition to domain organization, CBDs of GH6 and GH7 enzymes also showed variation in sequences. While the amino acids C11, G18, C22, and C28, according to the numbering of *T. reesei* CBH1, were completely conserved (Figure 5), the identity between the CBDs in GH6 and GH7 was only between 28 and 72% (Appendix A). Noteworthy variation in CBD sequences included Y, F, S, T, N at position 26, V, S, K, R, E, Y at position 30, and W, Y, F, I, C at position 34. In particular, the CBD in *V. mali* GH6 has a Y at both 30 and 34 positions, while the residue at 26 is replaced by S (Figure 5). Aromatic residues at these positions may contribute to substrate binding and thus CBH activities [60]. A comparative biochemical study of GH6 CBHs from different fungi has shown variation in substrate interactions, adsorption capacity, and turnover number [61]. Variation in CBD sequence has further been shown to influence the adsorption and kinetics of GH6 CBH [62]. Taken together, the results of this and previous studies indicate the potential role of cellulase sequence and domain organization in host preferences of pathogenic fungi [39,62,63,64].

The carbohydrate-active enzyme (CAzyme) profile of pathogenic fungi further reflects variation in chemical and structural properties of host PCWs [65]. A clustering analysis indicates a unique profile of CAZymes for the pathogenic fungi (Figure 6), such as *V. mali*, an apple-tree-specific pathogen [20,38]. Compared to herbaceous pathogens, *V. mali* contains relatively low copy numbers of AA3 and AA9 genes, but relatively high numbers of GH3 and GH43. AA9 along with AA3 are known to be involved in cellulose degradation by cellulolytic fungi [66], while GH3 and GH43 are involved in hemicellulose degradation or modification [67,68]. High levels of GH3 and GH43 in the *V. mali* genome may reflect their potential role in its hydrolytic preference for the AB substrate (Figure 1A). This is consistent with a decreased hydrolytic pereference of *V. mali* for the AB substrate after hemicellulase pretreatment (Figure 1B).

## 3. Materials and Methods

### 3.1. Chemicals and Materials

PCW biomass of wheat straw (WS), switchgrass straw (SW), rapeseed straw (RS), and apple tree branch (AB) were obtained in Yangling, Shaanxi, China. The biomass samples were cut into 1 cm using a chopper, soaked in tap water for one day, dried for 72 h at 50 °C in a drying oven, and homogenized for 2 min in a blender, followed by sieving through a 35-mesh screen. The sieved biomass was used for further experiments. The fungal strains used in this study were obtained from the College of Plant Protection and Shaanxi Key Laboratory of Molecular Biology for Agriculture, Northwest A&F University, Yangling, Shaanxi, China. Chemicals were purchased from Sigma-Aldrich (St. Louis, MO, USA) or Solarbio (Beijing, China).

### 3.2. Cultures and Growth Conditions

Spores or mycelia of fungal isolates were stored in 20% glycerol at −80 °C. When inoculum was required, the isolates were sub-cultured on potato dextrose agar (PDA) and grown at 25 °C, as previously described [39]. Then, 6 mm plugs of 7-day-old cultures were transferred to switchgrass-based agar media in 50 mm Petri dishes. The agar media were modified from the ATCC cellulose medium 907 (0.5 g (NH4)_2_SO_4_, 0.5 g L-asparagine, 1 g KH_2_PO_4_, 0.5 g KCl, 0.2 g MgSO_4_, 0.1 g CaCl_2_, 0.5 g yeast extract, 16 g agarose, 5 g dry switchgrass, 1 L H_2_O). Cultures were grown on the media for an additional 10 days before freezing at −80 °C. Exoproteome samples were extracted from the cultures and frozen at −80 °C until assayed as previously described [39]. 

### 3.3. Microscale Enzymatic Saccharification

The effectiveness of exoproteomes to hydrolyze raw and pretreated wheat straw, switchgrass, rapeseed straw, and apple tree branch biomass was assessed by enzymatic saccharification performed in 96-well flat-bottom microplates (Corning Life Sciences, Costar flat bottom 3370, Corning, NY, USA), as described previously [39,69]. Upon completion of saccharification, the samples were kept at −20 °C until further analysis. The total reducing sugars were quantified in a microplate format [69] using the DNS method, with glucose as the standard [70].

### 3.4. Hemicellulase and Peracetic Acid Pretreatment 

Hemicellulase H2125 (Sigma) was from *Aspergillus niger*, containing a mixture of xylanase, mannanase, and other activities. Briefly, air-dried biomass powder was treated with hemicellulases at 350 U/g of biomass for 72 h at 50 ± 2 °C or exoproteome of a pathogenic fungus at 200 mg/g of biomass for 12 h at 50 ± 2 °C in a sodium acetate buffer containing 0.05 g of dry biomass powder per ml of buffer. After the pretreatment, enzymes were inactivated by incubating at 100 °C for 10 min. Pretreated biomass was washed with 10 volumes of distilled water three times to remove residual sugars. Pretreated samples were air-dried and stored at room temperature for further experiments.

Peracetic acid (PPA) pretreatment was applied for selective delignification at a final loading of 5.0 g/g of biomass in a sodium acetate buffer containing 0.05 g of dry biomass powder per ml of buffer for 48 h at 25 ± 2 °C, as described previously [71]. Pretreated samples were neutralized by washing with distilled water. Pretreated samples were air-dried and stored at room temperature for further experiments.

### 3.5. Cell Wall Composition Analysis

Biomass samples were incubated in a hot air oven maintained at 105 °C until a constant weight was obtained. Extractive-based compositional analyses of the samples were performed according to the NREL LAPs [72,73]. Triplicate samples of approximately 0.3 g (dry basis) were weighed accurately into test tubes. 72% (*w*/*w*) H_2_SO_4_ was added to each of the tubes to make a sample to acid ratio of 1 g/10 mL. The tubes were placed in a water bath set at 30 °C for 1 h and were stirred in every 15 min with glass stirring rods. Each sample was diluted to 4% (*w*/*w*) H_2_SO_4_ with distilled water, followed by a secondary hydrolysis performed in an autoclave at 121 °C for 1 h (liquid setting). The samples were rapidly cooled to room temperature on ice, and then filtered through 0.22 μm MicroPES filters. The quantities of Klason lignin were determined gravimetrically [73]. Cellulose and hemicellulose contents were measured using an ICS 5000+ Ion Chromatography system equipped with a pulse-amperometric detector (Thermo Fisher Scientific, Waltham, MA, USA). A pre-flow of 200 mM NaOH for 15 min, followed by 12 mM NaOH for 15 min, was carried out to equilibrate a 4 × 250 mm Dionex CarboPac PA10 column with a 4 × 50 mm CarboPac PA10 BioLCTM guard column (Thermo Fisher Scientific) before injection of samples. A sample volume of 20 μL was injected into the column and eluted at 30 °C with 12 mM sodium hydroxide at a flow rate of 1 mL/min. Cellulose, hemicellulose, and lignin contents were calculated as described previously [73,74].

### 3.6. FESEM Imaging

To prepare the cell wall fragments for FESEM imaging, AB, RS, SG, or WS was washed, dried, and homogenized as described above. The derived biomass was treated with PAA, followed by washing with water and drying in a drying oven. Samples were placed on a carbon tape attached to stubs and coated with approximately 1 nm of iridium. FESEM images were taken using a FEI Nova Nano SEM 450 FEG SEM operated at 10.0 kV with a spot size of 3.0 as described in previous studies [75,76].

### 3.7. CAZyme Data Collection

CAZyme data of *F. graminearum, G. graminis*, *S. sclerotiorum*, *N. crassa* and *T. reesei* were from the reference [65] and CAZy database (http://www.cazy.org/, accessed on 16 November 2021). The data of *B. dothidea* were from the references [22,77]. The data of *B. sorokinina, C. gloeosporioides, R. cerealis,* and *V. mali* were from the references [38,78,79,80], respectively. 

### 3.8. GH6 and GH7 Gene Sequence Acquisition and Analysis

Sequences of GH6 and GH7 CBH genes in *T. reesei* were retrieved from the UniProt portal (www.uniprot.org, accessed on 16 November 2021); these sequences were used to Blast in the MycoCosm (https://mycocosm.jgi.doe.gov/, accessed on 16 November 2021) or EnsemblFungi (http://fungi.ensembl.org/, accessed on 16 November 2021) database to obtain GH6 and GH7 candidate genes coding sequences. The incomplete reading frame, short and redundant sequences were removed to obtain GH6 and GH7 genes and protein sequences. Multiple sequence alignments were conducted using MUSCLE [81] (https://www.ebi.ac.uk/Tools/msa/muscle/, accessed on 16 November 2021) and ESprint (http://espript.ibcp.fr/ESPript/cgi-bin/ESPript.cgi, accessed on 16 November 2021). The phylogenetic analysis was performed based on amino acid sequences using the maximum likelihood method in the Mega X program [82]. A total of 1000 replicates were used to test the reliability of the branches. ITOL was used to generate interactive phylogenetic tree [83].

### 3.9. Data Analysis and Statistics

All experiments were conducted with at least three biological replicates. Student’s *t*-test was used to analyze statistical significance between different variables. The heatmaps presented in Figure 1 and Figure 6 were generated using ClustVis (http://biit.cs.ut.ee/clustvis, accessed on 16 November 2021) [84].

## 4. Conclusions

Forest and agricultural residues are important renewable sources of biopolymers. In this study, the enzymatic digestibility of woody and herbaceous biomass by pathogenic fungi was compared before and after biomass pretreatment by hemicellulases and PAA. Preferred hydrolysis of apple tree branch (AB) (woody dicot), rapeseed straw (RS) (herbaceous dicot), or wheat straw (WS) (herbaceous monocot) were observed by *Valsa mali*, *Rhizoctonia cerealis*, or *Sclerotinia sclerotiorum*, respectively (Figure 1A). Hemicellulase pretreatment significantly changed hydrolytic preferences of the pathogenic fungi (Figure 1B). For instance, hemicellulase pretreatment switched the hydrolytic preference of *C. gloeosporioides* from AB to WS, while the pretreatment switched the preference of *G. graminis* from WS, SG, and RS to WS. Delignification by peracetic acid pretreatment enhanced the enzymatic digestibility of the biomass substrates tested (Figure 3D). It was noteworthy that the observed enhancement was generally more profound with non-host than host plant biomass. FESEM imaging showed different surface structures between the cell walls of apple tree and wheat or rapeseed straw, particularly in cellulose assembly and organization (Figure 4). The difference in woody and herbaceous cell wall structure was reflected by variation in PCWDEs of pathogenic fungi (Figure 5 and Figure 6). Relatively high abundance of GH3 and GH43 hemicellulase genes was found in the *V. mali* genome compared to that of the herbaceous pathogens tested (Figure 5A). The woody pathogens, such as *V. mali* and *Botryosphaeria dothidea*, appeared to have a unique organization of cellulose-binding domain (CBD) in GH6 and GH7 cellobiohydrolases. Taken together, the results of this and previous studies support the hypotheses that woody and herbaceous plant cell walls form different interaction networks of cellulose, hemicelluloses, and lignin, and that pathogenic fungi have evolved host-specific arsenals of PCWDEs, reflected by their genomic composition, domain organization, and sequences.

## Figures and Tables

**Figure 1 molecules-26-07220-f001:**
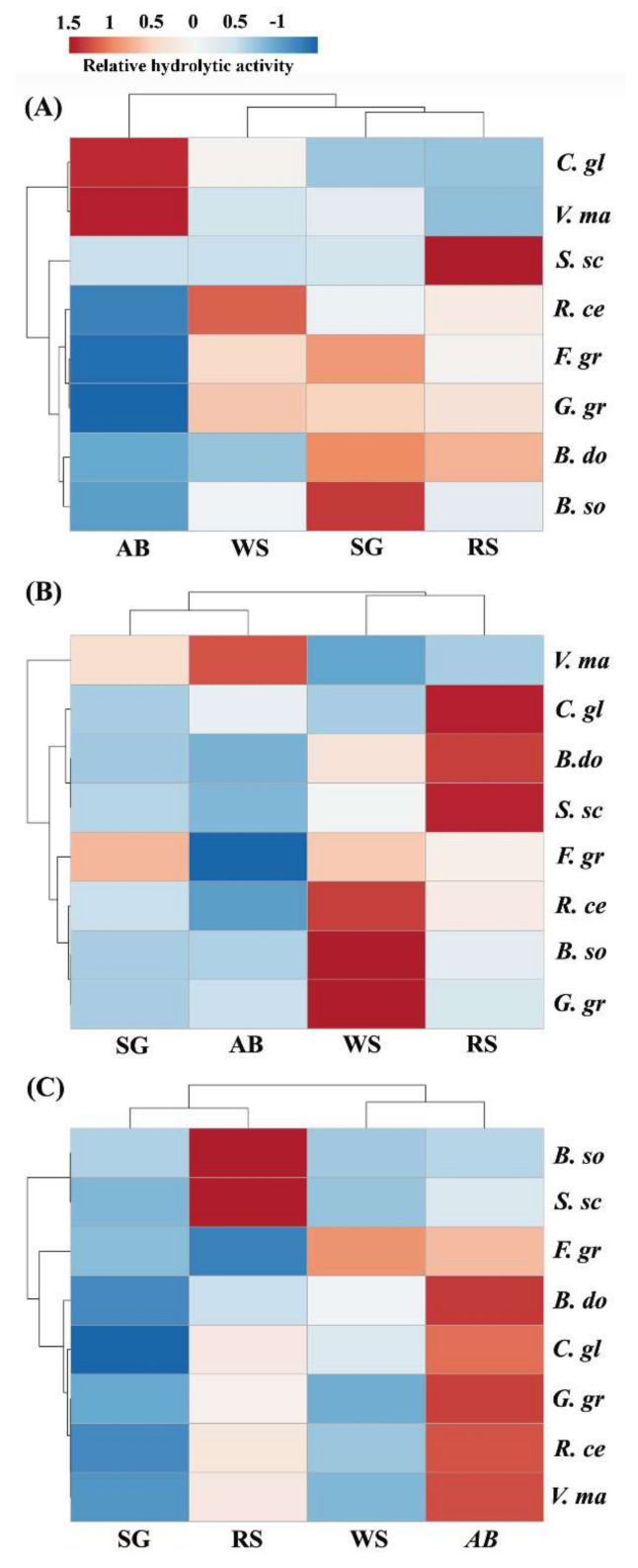
Hierarchical clustering of the pathogenic fungi tested. Heatmap showing the mean activities and clustering of 8 species of plant pathogenic fungi (*B. do*, *Botryosphaeria dothidea*; *B. so*, *Bipolaris sorokiniana*; *C. gl*, *Colletotrichum gloeosporioides*; *F. gr*, *Fusarium graminearum*; *G. gr*, *Gaeumannomyces graminis*; *S. sc*, *Sclerotinia sclerotiorum*; *V. ma*, *Valsa mali*), when assayed for hydrolysis of 4 crop residues (AB, apple tree branch; WS, wheat straw; SG, switchgrass straw; RS, rapeseed straw) untreated (**A**) or pretreated by hemicellulases (**B**) or PAA (**C**). The activity assay was performed as described in a previous study [39].

**Figure 2 molecules-26-07220-f002:**
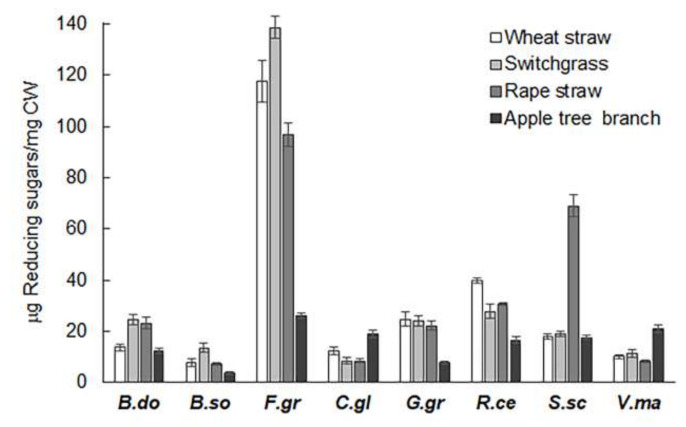
Enzymatic hydrolysis of cell wall substrates of wheat straw, switchgrass straw, rapeseed straw, and apple tree branch. Y axis depicts the amount of reducing sugars released after incubation of untreated cell wall substrates with extracts from the cultures of pathogenic fungi grown on switchgrass, as described previously [39].

**Figure 3 molecules-26-07220-f003:**
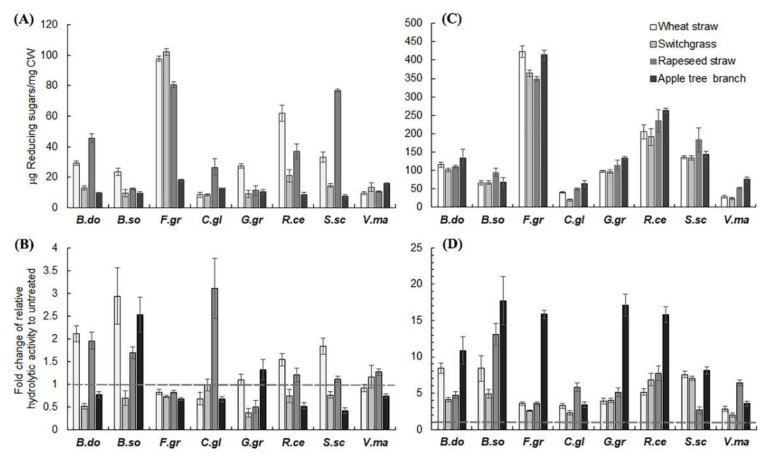
Enzymatic hydrolysis of cell wall substrates of wheat straw, switchgrass straw, rapeseed straw, and apple tree branch pretreated by hemicellulases (**A**) or PAA (**C**). (**B**,**D**) Fold-changes of hydrolytic activities of pathogenic fungi on hemicellulase- and PAA-pretreated substrates vs. untreated substrates, respectively.

**Figure 4 molecules-26-07220-f004:**
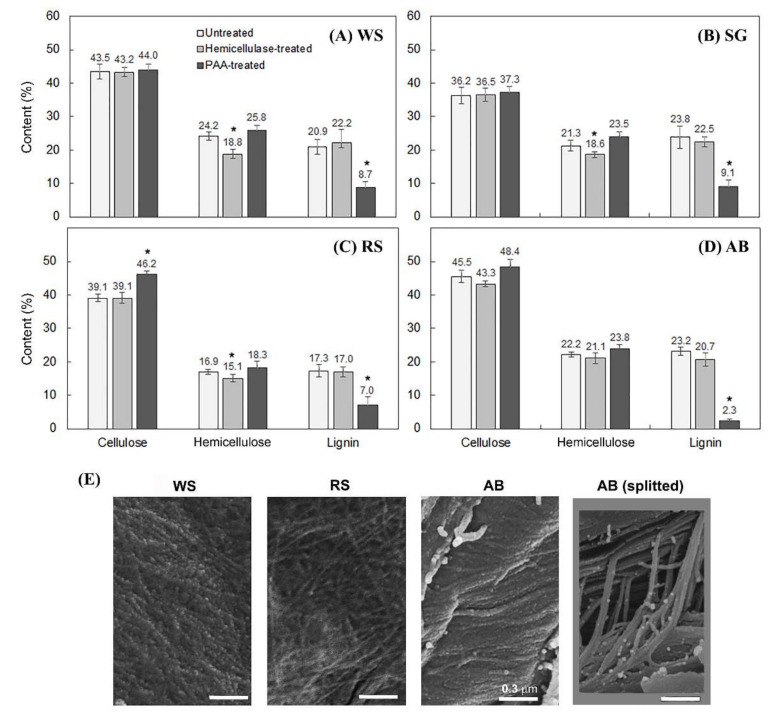
Compositional analysis and FESEM imaging of plant cell walls. Compositional analysis was performed with cell wall samples of (**A**) wheat straw (WS), (**B**) switchgrass straw (SG), (**C**) rapeseed straw (RS), and (**D**) apple tree branch (AB) before or after pretreatment by hemicellulases or PAA. Data are the means of three biological replicates and error bars show the standard deviation. Asterisks indicate the significant difference (determined by *t*-test) between hemicellulase-pretreated or PAA-pretreated and untreated crop residue biomass (* *p* < 0.01). (**E**) FESEM images were obtained after PAA pretreatment to expose fibrillar cellulose on cell wall surfaces. Scale bar, 0.3 μm.

**Figure 5 molecules-26-07220-f005:**
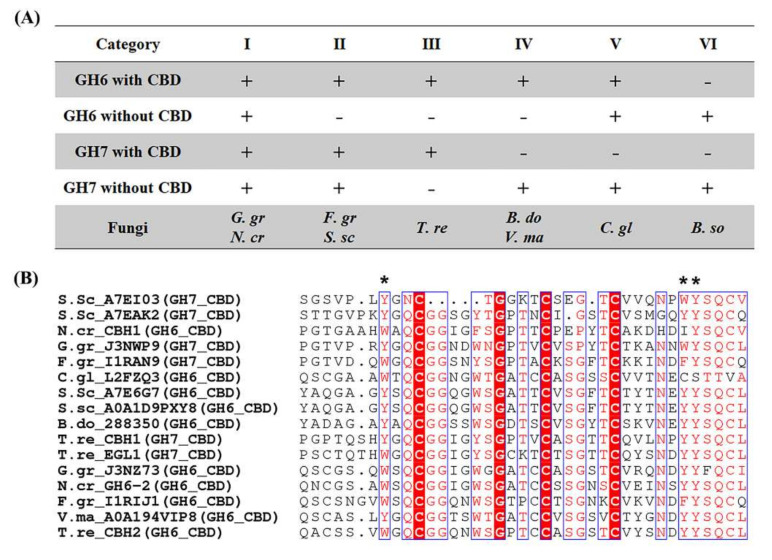
CBD organization and sequence alignment. (**A**) Categorization of CBD organization in GH6 and GH7 cellulases from the pathogenic fungi tested. + and − indicate cellulases with and without CBD, respectively. (**B**) Sequence alignment of CBDs in GH6 and GH7 enzymes. Red color highlights conserved residues. Black stars (*) are aromatic residues in the planar surface of CBD involved in cellulose binding.

**Figure 6 molecules-26-07220-f006:**
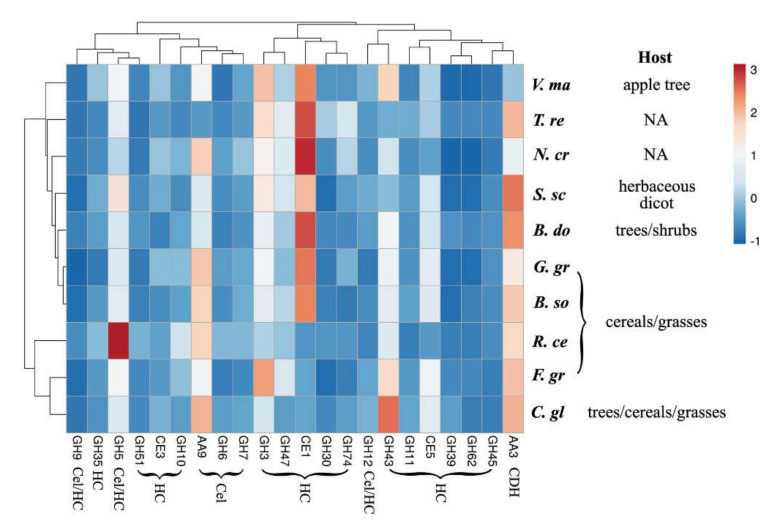
Hierarchical clustering of the pathogenic fungi tested, based on the abundance and composition of CAZyme classes. Heatmap shows the relative abundance of each CAZyme class. CAZymes categories include Glycoside Hydrolases (GHs), Carbohydrate Esterases (CEs), and Auxiliary Activities enzymes (AAs). Catalytic activities of the CAZymes include cellulase (Cel), hemicellulase (HC), and cellobiose dehydrogenase (CDH). The AA9 family consists of lytic polysaccharide monooxygenases (LPMOs) involved in cellulose degradation. A detailed description of the CAZymes is available on the CAZy database (http://www.cazy.org, accessed on 16 November 2021).

**Table 1 molecules-26-07220-t001:** Fungal pathogens used in this study.

Pathogenic Species	Phylum	Host	Disease	Reference
*Bipolaris sorokiniana*	Ascomycota	Cereals (e.g., wheat) and grasses (e.g., switchgrass).	Causes disease on the root, leaf and stem, and head tissue.	[25,26]
*Fusarium graminearum*	Ascomycota	Cereals (e.g., wheat) and grasses (e.g., switchgrass)	Causes Fusarium head blight and Gibberella ear rot and stalk rot.	[28,29,30]
*Gaeumannomyces graminis*	Ascomycota	Cereals (e.g., wheat) and grasses.	Colonizes the root and crown tissue, causing Turfgrass disease.	[27]
*Rhizoctonia cerealis*	Basidiomycota	Cereals (e.g., wheat) and grasses (e.g., switchgrass)	Causes sharp eyespot and root rot.	[31,32]
*Sclerotinia sclerotiorum*	Ascomycota	Dicotyledonous herbaceous species (e.g., rapeseed, soybean)	Causes Sclerotinia head rot, Sclerotinia stalk rot, and Sclerotinia wilt.	[24]
*Valsa mali var. mali*	Ascomycota	Apple tree	Preferentially infects apple trees, causing canker diseases.	[20,38]
*Botryosphaeria dothidea*	Ascomycota	Trees and shrubs (e.g., apple and other fruit trees)	Disease symptoms are associated with twig, branch and stem cankers, tip and branch dieback, fruit rot, etc.	[21,22]
*Colletotrichum gloeosporioides* (teleomorph: *Glomerella cingulata*)	Ascomycota	Trees (e.g., apple tree), cereals and grasses, legumes, vegetables	Causes anthracnose disease	[23]

## Data Availability

Data is contained within the article.

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
