# Peer review of "Comparative Analysis of Herbaceous and Woody Cell Wall Digestibility by Pathogenic Fungi"

_molecules, 2021, doi:10.3390/molecules26237220_

Round 1
Reviewer 1 Report
Biofuels obtained from plant material are environmentally friendly alternatives to fossil fuels. One of the major limitation of the production of plant based biofuels is the recalcitrance of the plant cell wall due to compounds such as hemicellulose and lignin, that seems to act as a physical barrier to microbial enzymes that can/could produce fermentable sugars from plant material. Dou et al. carried out multiple experiments to compare the structure and digestibility of herbaceous and woody plant cell wall material. They focused on the digestibility of the substrate by fungal exoproteomes obtained from different plant pathogenic fungi including specialist pathogens and pathogens with a wide host range. They have found that the enzyme mixtures of the pathogenic fungi show the most profound hydrolytic activity on the substrates that were obtained from their host plant. They have also compared the digestibility of these substrate following hemicellulase and peracetic acid pretreatment and found significant differences between the digestibility of treated and untreated samples. They have also investigated the surface of the plant cell wall material using FESEM microscopy and compared the cell wall composition of the herbaceous and woody plant cell wall material. In addition they have investiated the CAZy arsenal f the selected plant pathogens.
Major comments
By itself, the data obtained from the measurement of hydrolytic activity of the selected plant pathogenic fungi on mococot grass, dicot grass and dicot woody plants before and after pretreatment can be interesting for a specific group within the community of resarchers interested in plant cell wall degradation/biofuel production. However, I have an overall feeling that the authors failed to interpret these results correctly. Please see some specific comments on this below.
- How is it possible that both hemicellulase and peracetic acid treatment changed the substrate preference of several investigated species? The interpretation of this interesting result is completely lacking.
2.From line 145 authors discuss the cell wall composition of the selected plant materials without putting it into context. What is considered as high level of high level of hemicellulose? How these data relate to the data in the literature?
- The changes of the cell wall compositions of the plant substrates are less profound after hemicellulase treatment than after peracetic acid treatment. I did not find any data on the activity of this particular hemicellulase. Does this change in hemicellulose content align well with similar data in the literature?
- In my opinion the weakest point of the manuscript is the FESEM analyis. The description of the method is far from sufficient? E.g. how were the samples prepared? Also, in this form, with these 4 bad quality image and the limited amount of information concluded from them it is completely unnecessary and says nothing new to the field. In my opinion this part should be excluded or interpreted more thouroughly.
- I find the CAZy analysis very superficial. Plant pathogenic fungi has a plethora of carbohydrate active enzymes. Selecitng GH6 and GH7 families and suggesting a relationship between the difference in the cell wall composition of host plants and the domain composition of these selected families is not justified at all. Also, Figure 6 shows the abundance of CAZy of the investigated species. Why did T. reesei and N. crassa become involved in this analysis? Are these the full CAZy repertoire of the selected species? The description of the CAZy annotation is quite poor in the Methods section, a Supplementary table would be really useful with the proper references. In addition, in my opinion, a heatmap showing copy numbers of the CAZy families would be more informative than a hierarchical clustering of the species based on the abundance of the CAZy families.
Minor comments.
Italicize species names in the main text
Line 27: „...woody and herbaceous plant cell walls form different but unique interaction networks...” -> Repharse What does „different but unique” supposed to mean?
Line 34 – 40: This paragraph is basically plagiarized from the introduction of reference 5 (Ding et al., 2012) and needs to be rephrased!
Line 43-46 – This was also copied and only moderatley from the introduction of Patthatil et al., 2015.
Line 57: - 1,4 linked -> missing β.
Line 64-72: These sentences suggests that comparisons were made within subgroups (e.g.between only woody plant pathogens.) Please make it clear that the enzymatic activity of each observed species was measured on each subtrate.
Line 75- fingal > fungal
Line 128 -> mixture of xylanases and other activites -> rephrase
Line 164
Line 172 Choi et al. -> correct citation format
Figure 1. I assume that the legend at the upper left corner of Figure1 shows Z-score values, but it is not indicated. Please indicate „Z-score” on the figure legened. Also, instead of column–wise oredring of the heatmaps, please consider a rowwise ordering-
Line 205-206: It is unnecessary to write it down again, that wheat straw is herbaceous monocot etc.
Line 213 -> Figure4D shows the compositional analysis of the apple tree branch, not a FESEM image.
Line 215 - > Donaldson 2007. This is lacking. from the reference list, also, correct the citation format.
Line 237-238. -> What does suppose to mean?: „...their relative abundance [CBH1&CBH2] is 60% and 20% respectively. Please clarify.
Descritption of the FESEM data and CAZyme data collection is very superficial and far from reproducibility. Please rewrite them!
I found no reference for Table S1 and Video S1 in the main text.
Author Response
Manuscript ID: molecules-1453707
Title: Comparative analysis of herbaceous and woody cell wall digestibility by pathogenic fungi
The following is a point-by-point response to reviewer’s comments and concerns.
Reviewer 1
- How is it possible that both hemicellulase and peracetic acid treatment changed the substrate preference of several investigated species? The interpretation of this interesting result is completely lacking.
Author response: Thank you for pointing this out. We have revised the manuscript accordingly and the revised parts are highlighted by yellow background color.
Recent studies suggest that plant cell wall (PCW) recalcitrance to enzymatic hydrolysis may be the result of inherent limitations in cell wall porosity, the fine structure of PCW polymers, and cellulose features such as crystallinity. Therefore, it is critical to understand how PCW polymers and their assembly influence PCW recalcitrance or degradability. However, due to the high complexity of PCW composition and structure, it remains a challenging task to investigate the relationships among PCW polymers, their assembly and recalcitrance to enzymatic hydrolysis. This study provided a strategy to examine the above relationships by using pathogenic fungi and their host biomass substrates. Preferred hydrolysis of apple tree branch (woody dicot), rapeseed straw (herbaceous dicot), and wheat straw (herbaceous monocot) was observed by the apple tree-specific pathogen Valsa mali, the herbaceous monocot pathogen Rhizoctonia cerealis, and the herbaceous dicot pathogen Sclerotinia sclerotiorum, respectively, reflecting differences of the cell wall substrates of the host plants and host-specific plant cell wall-degrading enzyme (PCWDE) cocktails produced by the plant pathogens. Delignification by peracetic acid pretreatment increased digestibility of the substrates and the increase was generally more profound with non-host than host plant biomass substrates, confirming the idea that lignin is a major contributor to PCW recalcitrance to enzymatic hydrolysis. Hemicellulase pretreatment significantly changed hydrolytic preferences of the selected pathogens for the biomass substrates, further indicating the importance of hemicelluloses in plant cell wall assembly and their impacts on the access of lignocellulases to cellulose microfibrils or macrofibrils.
- From line 145 authors discuss the cell wall composition of the selected plant materials without putting it into context. What is considered as high level of high level of hemicellulose? How these data relate to the data in the literature?
Author response: Thank you for pointing this out. We have deleted the sentence “Cell wall composition analysis showed a medium level of hemicelluloses in AB and SG (Figure 4B), but a relatively high level in WS (Figure 4A) and a low level in RS (Figure 4C)” and replaced it by “The composition of the crop residues was also determined (Figure 4) and the derived contents of cellulose, hemicellulose, and lignin were in the range reported in previous studies (Alemdar & Sain, 2008; Jiang et al., 2018; Karagöz et al., 2012; Liu et al., 2013; Lu et al., 2009; Lyu et al., 2017; Qiu et al., 2018; Xu et al., 2011; Zhang et al., 2013a)”. The revised parts are highlighted by yellow background color in the revised manuscript.
PCW composition depends on species, environmental conditions, developmental stage and cell type. The derived contents of hemicelluloses as well as cellulose and lignin were in the range reported in previous studies in wheat straw (References: Alemdar, A., Sain, M. 2008. Isolation and characterization of nanofibers from agricultural residues - wheat straw and soy hulls. Bioresource Technology, 99(6), 1664-1671; (2) Jiang, B., Yu, J., Luo, X., Zhu, Y., Jin, Y. 2018. A strategy to improve enzymatic saccharification of wheat straw by adding water-soluble lignin prepared from alkali pretreatment spent liquor. Process biochemistry, 71, 147-151; (3) Qiu, J., Tian, D., Shen, F., Hu, J., Zeng, Y., Yang, G., Zhang, Y., Deng, S., Zhang, J. 2018. Bioethanol production from wheat straw by phosphoric acid plus hydrogen peroxide (PHP) pretreatment via simultaneous saccharification and fermentation (SSF) at high solid loadings. Bioresource Technology, 268, 355-362;), rapeseed straw (Reference: Karagöz, P., Rocha, I.V., Özkan, M., Angelidaki, I. 2012. Alkaline peroxide pretreatment of rapeseed straw for enhancing bioethanol production by same vessel saccharification and co-fermentation. Bioresource Technology, 104, 349-357), switchgrass (References: (1) Liu, J., Wang, M.L., Tonnis, B., Habteselassie, M., Liao, X., Huang, Q. 2013. Fungal pretreatment of switchgrass for improved saccharification and simultaneous enzyme production. Bioresource Technology, 135, 39-45; (2) Liu, J., Wang, M.L., Tonnis, B., Habteselassie, M., Liao, X., Huang, Q. 2013. Fungal pretreatment of switchgrass for improved saccharification and simultaneous enzyme production. Bioresource Technology, 135, 39-45; (3) Xu, B., Escamilla‐Treviño, L.L., Sathitsuksanoh, N., Shen, Z., Shen, H., Percival Zhang, Y.-H., Dixon, R.A., Zhao, B. 2011. Silencing of 4-coumarate:coenzyme A ligase in switchgrass leads to reduced lignin content and improved fermentable sugar yields for biofuel production. New Phytologist, 192(3), 611-625; (4) Zhang, D., Yang, Q., Zhu, J., Pan, X. 2013a. Sulfite (SPORL) pretreatment of switchgrass for enzymatic saccharification. Bioresource Technology, 100(129), 127-134), and apple tree (Reference: Lyu, S., Lyu, D., Du, G., Yang, Y. 2017. Apple Branch Decomposition and Nutrient Turnover in the Orchard Soil. BioResources, 12(2), 3108-3121). For instance, in hardwoods (dicotyledonous species), hemicelluloses contribute to 19-34% of the dry weight (Reference: Sjöström, E. 1993. Wood Chemistry - Fundamentals and Applications. San Diego, CA: Academic Press). In apple tree, hemicellulose content was found to be 22.2%, which fall into the range in hardwoods.
- In my opinion the weakest point of the manuscript is the FESEM analysis. The description of the method is far from sufficient? E.g. how were the samples prepared? Also, in this form, with these 4 bad quality image and the limited amount of information concluded from them it is completely unnecessary and says nothing new to the field. In my opinion this part should be excluded or interpreted more thouroughly.
Author response: We agree with reviewer’s opinion. As suggested, additional description of the method as well as discussion of the results were included. The revised parts are highlighted by yellow background color in the revised manuscript. Our previous study using fluorescence quenching assay revealed that mutations in cellulose synthases affect the nano-scale porosity of plant cell walls (Reference: Liu, X.; Li, J.; Zhao, H.; Liu, B.; Günther-Pomorski, T.; Chen, S.; Liesche, J., Novel tool to quantify cell wall porosity relates wall structure to cell growth and drug uptake. J Cell Biol 2019, 218, (4), 1408-1421). Though the same assay could not be used to quantify the porosity of the secondary wall of crop residues, FESEM imaging was applied to examine if cellulose organization varies in different plant species. To prepare the cell wall fragments for FESEM imaging, biomass samples were cut into 1 cm using a chopper, soaked in tap water for one day, dried for 72 hours at 50°C in a drying oven, and homogenized for 2 min in a blender, followed by sieving through a 35-mesh screen. PAA treatment was applied to remove surface lignin to expose surface fibrillar cellulose. Sequence variation has been reported in CESAs of different plant species. Post-translational modifications such as phosphorylation of CESAs have been shown to modulate cellulose synthesis and deposition. Variation in CESA sequences and post-translational modifications may partially account for different cellulose structure and organization observed in different plant species, leading to different porosity of plant cell walls or cellulose accessibility by cellulolytic enzymes.
- I find the CAZy analysis very superficial. Plant pathogenic fungi has a plethora of carbohydrate active enzymes. Selecitng GH6 and GH7 families and suggesting a relationship between the difference in the cell wall composition of host plants and the domain composition of these selected families is not justified at all. Also, Figure 6 shows the abundance of CAZy of the investigated species. Why did T. reesei and N. crassa become involved in this analysis? Are these the full CAZy repertoire of the selected species? The description of the CAZy annotation is quite poor in the Methods section, a Supplementary table would be really useful with the proper references. In addition, in my opinion, a heatmap showing copy numbers of the CAZy families would be more informative than a hierarchical clustering of the species based on the abundance of the CAZy families.
Author response: Thank you for pointing this out. We have revised the manuscript accordingly and the revised parts are highlighted by yellow background color.
Our analysis confirmed the importance of lignin and hemicelluloses in PCW structure, which may partially account for the different degradability of host plant biomass by pathogenic fungi. In addition to hemicelluloses and lignin, cellulose crystallinity is considered another feature of plant cell wall that determines plant cell wall degradability. Our current and previous studies further indicate that cellulose deposition also play a role in modulating plant cell wall structure and degradability, as reflected by that mutations in CESAs modulate plant cell wall porosity, a critical parameter that determines cellulose accessibility to cellulolytic enzymes. To examine if variation in cellulose synthesis and deposition in different host plants is reflected by the co-evolved cellulases of pathogenic fungi, we examined the organization of cellulose-binding domains (CBDs) in the cellulases of the pathogenic fungi tested. Variation in CBD organization and sequences has been shown to influence both cellulase adsorption to cellulose and hydrolytic activities of cellulases. In this study, we did find variation in CBD sequences and organization in GH6 and GH7 cellulases of pathogenic fungi, corroborating with the idea that cellulose synthesis and deposition vary in different plant species, which may also contribute to plant cell wall degradability through effects on plant cell wall structure and porosity. However, further studies are needed to investigate the co-evolution relationships between host plant cellulose and cellulases of pathogenic fungi.
Detailed CAZy annotation is available in CAZy database (http://www.cazy.org).
- Minor comments. Italicize species names in the main text
Author response: Thank you for pointing this out. We have made revisions as suggested and highlighted the revisions by yellow background color.
- Line 27: „...woody and herbaceous plant cell walls form different but unique interaction networks...” -> Repharse What does „different but unique” supposed to mean?
Author response: We have revised this part and highlighted the revisions by yellow background color.
- Line 34 – 40: This paragraph is basically plagiarized from the introduction of reference 5 (Ding et al., 2012) and needs to be rephrased!
Author response: We have revised this part and highlighted the revisions by yellow background color.
- Line 43-46 – This was also copied and only moderatley from the introduction of Patthatil et al., 2015.
Author response: We have revised this part and highlighted the revisions by yellow background color.
- Line 57: - 1,4 linked -> missing β.
Author response: We have revised it.
- Line 64-72: These sentences suggests that comparisons were made within subgroups (e.g.between only woody plant pathogens.) Please make it clear that the enzymatic activity of each observed species was measured on each subtrate.
Author response: As suggested, the first sentence of this paragraph was replaced by “Effects of hemicellulase and PAA pretreatment on host substrate preferences were further compared with their pathogenic fungi (Table 1)”.
- Line 75- fingal > fungal
Author response: We have revised it.
- Line 128 -> mixture of xylanases and other activites -> rephrase
Author response: We have revised it.
- Line 172 Choi et al. -> correct citation format
Author response: We have revised it.
- Figure 1. I assume that the legend at the upper left corner of Figure1 shows Z-score values, but it is not indicated. Please indicate „Z-score” on the figure legened. Also, instead of column–wise oredring of the heatmaps, please consider a rowwise ordering-
Author response: The legend at the upper left corner of Figure 1 shows relative activities of the extracts from pathogenic fungi to hydrolyze the host plant biomass tested. It has been indicated as “relative hydrolytic activity”.
- Line 205-206: It is unnecessary to write it down again, that wheat straw is herbaceous monocot etc.
Author response: This sentence has been deleted in the revised manuscript.
- Line 213 -> Figure4D shows the compositional analysis of the apple tree branch, not a FESEM image.
Author response: Thank you for pointing this out. It has been revised as Figure 4E.
- Line 215 - > Donaldson 2007. This is lacking. from the reference list, also, correct the citation format.
Author response: This sentence has been deleted in the revised manuscript.
- Line 237-238. -> What does suppose to mean?: „...their relative abundance [CBH1&CBH2] is 60% and 20% respectively. Please clarify.
Author response: This sentence has been revised as “their relative abundance is 60% and 20% in the exoproteome of T. reesei, respectively”
- Descritption of the FESEM data and CAZyme data collection is very superficial and far from reproducibility. Please rewrite them!
Author response: The relevant parts have been revised as suggested and are highlighted by yellow background color.
CAZyme data of F. graminearum, G. graminis, S. sclerotiorum, N. crassa and T. reesei were from the reference [38] and CAZy database (http://www.cazy.org/). The data of B. dothidea were from the references [39] and [17]. The data of B. sorokinina, C. gloeosporioides, R. cerealis, and V. mali were from the references [40], [41], [42], and [43], respectively.
- I found no reference for Table S1 and Video S1 in the main text.
Author response: No Table S1 and Video S1 were included in the original manuscript

Reviewer 2 Report
I would like to express my thanks to you due to the idea of the research and techniques applied. I just have few comments as follows:
1- Line 127, the name of taxon should be itilacized.
2-Please add the name authority of all taxa mentioned in table 1
3- taxon in line 367 should be italicaized
Author Response
Reviewer 2
- I would like to express my thanks to you due to the idea of the research and techniques applied. I just have few comments as follows. Line 127, the name of taxon should be itilacized.
Author response: Thank you for pointing this out. We have revised the manuscript as suggested and the revised parts are highlighted by yellow background color.
- Please add the name authority of all taxa mentioned in table 1
Author response: Thank you for pointing this out. We revised the names as suggested.
- taxon in line 367 should be italicaized
Author response: Thank you for pointing this out. We have revised it as suggested.

Reviewer 3 Report
Title: Comparative analysis of herbaceous and woody cell wall digestibility by pathogenic fungi
Authors: Dou et al.
The authors compared the enzymatic digestibility of selected woody and herbaceous biomass by pathogenic fungi. Although the study is well designed and interesting to the reader, the manuscript requires several revisions according to the comments below.
Title:
Why do the authors point to the keyword "pathogenic"? What is the significant problem of present research?
Abstract:
-Make sure that all abbreviations have been defined when first used. e.g. FESEM, CAZ, GH, PCWDEs etc.
-What is experimental design?
-There is a lack of a summary of the methodology in this section.
-Section of the short conclusion needs to focus on the objective study. Add it.
Introduction:
L41-42: " few studies have compared the enzymatic digestibility." Do the authors surely know that there are a few?
L43-45: The sentence requires citation!
L44: The authors mention "biomass feedstocks", but how are the keywords "pathogenic" harmful to animal feed?
L53-55: How did the authors select those materials to evaluate?
L56-57: Why? Create a biological mechanism.
L64: Why are pathogenic fungi of interest to the authors? Specifics
L65-72: There are several pathogenic fungi that were tested and successfully reported. Thus, the present study did not clarify what the new knowledge obtained would be, and the research gap was not well addressed.
-Hypothesis is lacking from the study.
Results and Discussion
This section as a whole lacks a thorough explanation of the mechanism biology of the interim influence treatment study. The authors are required to add more details. In addition, check the order No of citations throughout the manuscript.
L83-85: Remain not clear why those plant materials were chosen. What is the research gap?
L133: What is the meaning of "crop residue digestibility" that the authors intend to detail? This keyword should be mentioned in the abstract/introduction as well.
L172: L216: Check citation style!
L247-248: How could the authors assume this statement? Describe your experience!
Materials and Methods
L276-278: How were those plants' materials chosen?
Conclusion:
The too-long conclusion here! The authors should focus on what research objective established and how are the key result could reply to it. Also, this section did not well respond to the problem as indicated in the section of the introduction. The present form is not attractive to the reader to gain the key message from the authors. Referring to all these figures here should be removed.
Reference:
-Remove the old citation and replace the new one. Older than 2010 should be removed.
-Check format according to journal guidelines.
Author Response
Reviewer 3
- The authors compared the enzymatic digestibility of selected woody and herbaceous biomass by pathogenic fungi. Although the study is well designed and interesting to the reader, the manuscript requires several revisions according to the comments below. Title: Why do the authors point to the keyword "pathogenic"? What is the significant problem of present research?
Author response: Thank you for your review. To deconstruct host plant cell walls, pathogenic fungi have evolved plant cell wall-degrading enzymes (PCWDEs). This study examined whether and how enzyme extracts from pathogenic fungi differentially hydrolyze plant cell wall substrates from different host plants, which helps understand the relationship of host plant cell wall structure and enzymatic digestibility. Greater understanding of enzymatic digestibility of forest and agricultural biomass is critical for enabling cost-effective production of sustainable biofuels and bioproducts as low efficiency of this process is still a major challenge for the modern biorefinery industry.
We have revised the manuscript as suggested and the revised part in the abstract is highlighted by yellow background color.
- Abstract: -Make sure that all abbreviations have been defined when first used. e.g. FESEM, CAZ, GH, PCWDEs etc.
Author response: Thank you for pointing this out. We have revised the abstract as suggested.
- What is experimental design?
Author response: in this manuscript, we compared the enzymatic digestibility of selected woody and herbaceous biomass by pathogenic fungi. To further examine the role of lignin and hemicelluloses in biomass digestibility, biomass substrates were pretreated by peracetic acid (PAA) and hemicellulases, respectively. We observed preferred hydrolysis of apple tree branch (woody dicot), rapeseed straw (herbaceous dicot), or wheat straw (herbaceous monocot) by the apple tree-specific pathogen Valsa mali, the herbaceous monocot pathogen Rhizoctonia cerealis, or the herbaceous dicot pathogen Sclerotinia sclerotiorum, respectively. Delignification by PAA increased digestibility of PCW substrates and the increase was generally more profound with non-host than host PCW substrates. Hemicellulase pretreatment only slightly reduced or had no effect on hemicellulose contents in the PCW substrates, indicating modification instead of removal of hemicelluloses; however, the pretreatment significantly changed hydrolytic preferences of the selected pathogens, reflecting a role of hemicellulose branching in PCW digestibility. Cellulose organization appear also to impact digestibility of host PCWs, as reflected by differences in cellulose microfibril organization in woody and herbaceous PCWs and variation in cellulose-binding domain organization in cellulases of pathogenic fungi, which is known to influence enzyme access to cellulose. Taken together, this study highlighted the importance of cellulose, hemicelluloses and lignin in host PCW architecture and enzymatic digestibility by fungal pathogens.
We have revised the abstract accordingly.
- There is a lack of a summary of the methodology in this section.
Author response: A summary of the methodology has been included in the revised abstract: “here, an approach integrating enzyme activity assay, biomass pretreatment, field emission scanning electron microscopy (FESEM), and genomic analysis of PCWDEs was applied to examine digestibility of selected woody and herbaceous biomass by pathogenic fungi”.
- Section of the short conclusion needs to focus on the objective study. Add it.
Author response: A short conclusion has been included in the revised manuscript: “Taken together, this study highlighted the importance of cellulose, hemicelluloses, and lignin in host PCW architecture and digestibility by fungal pathogens.”
- Introduction: L41-42: " few studies have compared the enzymatic digestibility." Do the authors surely know that there are a few?
Author response: This sentence has been revised as “Despite great progress in investigating plant cell wall recalcitrance, few studies have compared the enzymatic digestibility of herbaceous and woody biomass by pathogenic fungi”
- L43-45: The sentence requires citation!
Author response: Thank you for pointing this out. A citation has been included.
- L44: The authors mention "biomass feedstocks", but how are the keywords "pathogenic" harmful to animal feed?
Author response: Pathogenic fungi, such as Valsa mali, are capable of specifically targeting apple tree biomass, an abundant plant biomass resource in certain areas such as Shaanxi province, China. Examining the relationship between host plant cell walls and host-specific fungal pathogens helps understand how pathogens attack host plants. It may also help provide a strategy to improve plant cell wall digestibility and/or lignocellulase cocktails.
- L53-55: How did the authors select those materials to evaluate?
Author response: More information has been included in the revised manuscript: “In addition to the promising energy crop switchgrass (SG), wheat straw (WS) and rapeseed straw (RS) were used as representative residues of major monocot and dicot crops in Northwest China [17, 18]. Wood branches were rich in certain areas and apple tree branch (AB) was selected as it has the largest production in non-forest areas in Northwest China [19].”
- L56-57: Why? Create a biological mechanism.
Author response: This sentence has been revised as “Hemicellulase pretreatment makes changes to not only the main chains of polysaccharides but also branching decoration with sugars (i.e. mono- or oligosaccharides) or non-sugars (i.e. acetylation, methylation, or feruloylation) in the main chains [15]”.
- L64: Why are pathogenic fungi of interest to the authors? Specifics
Author response: More information has been included in the revised manuscript: “In addition to the promising energy crop switchgrass (SG), wheat straw (WS) and rapeseed straw (RS) were used as representative residues of major monocot and dicot crops in Northwest China [16, 17]. Wood branches were rich in certain areas and apple tree branch (AB) was selected as it has the largest production in non-forest areas in Northwest China [18]. Valsa mali, an apple tree-specific pathogen [20], was used as a woody plant pathogen. Botryosphaeria dothidea, which infects various woody species including apple tree [21, 22], was used as a comparison to the apple tree-specific V. mali. Another comparison was made with Colletotrichum gloeosporioides, which infects a broad range of host plants, including woody and herbaceous plants [23]. Sclerotinia sclerotiorum was used as a representative of herbaceous dicot pathogen that infects rapeseed [24], while Bipolaris sorokiniana [25, 26], Gaeumannomyces graminis [27], Fusarium graminearum [28-30], and Rhizoctonia cerealis [31, 32] were used as herbaceous monocot pathogens that infects wheat.”.
- L65-72: There are several pathogenic fungi that were tested and successfully reported. Thus, the present study did not clarify what the new knowledge obtained would be, and the research gap was not well addressed. Hypothesis is lacking from the study.
Author response: The following information has been included: “Comparing the hydrolytic activities of the selected pathogens would help determine whether the pathogens have host preferences. Pretreatment by PAA and hemicellulases would help further examine if lignin and hemicelluloses have impact on digestibility of host PCWs by pathogenic fungi. Our previous studies suggest that mutations in cellulose synthases influence not only cellulose synthesis and deposition [33, 34] but also PCW porosity [35], a critical parameter affecting the accessibility of cellulose to cellulolytic enzymes [36]”.
- Results and Discussion: This section as a whole lacks a thorough explanation of the mechanism biology of the interim influence treatment study. The authors are required to add more details. In addition, check the order No of citations throughout the manuscript. L83-85: Remain not clear why those plant materials were chosen. What is the research gap?
Author response: The relevant information has been included in the introduction as suggested: “In addition to the promising energy crop switchgrass (SG), wheat straw (WS) and rapeseed straw (RS) were used as representative residues of major monocot and dicot crops in Northwest China [17, 18]. Wood branches were rich in certain areas and apple tree branch (AB) was selected as it has the largest production in non-forest areas in Northwest China [19]. Valsa mali, an apple tree-specific pathogen [20], was used as a woody plant pathogen. Botryosphaeria dothidea, which infects various woody species including apple tree [21, 22], was used as a comparison to the apple tree-specific V. mali. Another comparison was made with Colletotrichum gloeosporioides, which infects a broad range of host plants, including woody and herbaceous plants [23]. Sclerotinia sclerotiorum was used as a representative of herbaceous dicot pathogen that infects rapeseed [24], while Bipolaris sorokiniana [25, 26], Gaeumannomyces graminis [27], Fusarium graminearum [28-30], and Rhizoctonia cerealis [31, 32] were used as herbaceous monocot pathogens that infects wheat”.
More explanation of the mechanism biology has also been included in the results and discussion and the revisions are highlighted by yellow background color.
- L133: What is the meaning of "crop residue digestibility" that the authors intend to detail? This keyword should be mentioned in the abstract/introduction as well.
Author response: The keyword was introduced in the title and abstract and was described in the introduction as well.
- L172: L216: Check citation style!
Author response: Thank you for pointing these out. We have made revisions as suggested.
- L247-248: How could the authors assume this statement? Describe your experience!
Author response: GH6 and GH7 of V. mali and B. dothidea can be categorized into group 5, in which GH6 enzymes contain a CBD but GH7 enzymes do not have a CBD (Figure S1).
- Materials and Methods. L276-278: How were those plants' materials chosen?
Author response: The relevant information has been included in the introduction as suggested: “In addition to the promising energy crop switchgrass (SG), wheat straw (WS) and rapeseed straw (RS) were used as representative residues of major monocot and dicot crops in Northwest China [17, 18]. Wood branches were rich in certain areas and apple tree branch (AB) was selected as it has the largest production in non-forest areas in Northwest China [19].”
This paragraph has been revised and revisions are highlighted by yellow background color.
- Conclusion: The too-long conclusion here! The authors should focus on what research objective established and how are the key result could reply to it. Also, this section did not well respond to the problem as indicated in the section of the introduction. The present form is not attractive to the reader to gain the key message from the authors. Referring to all these figures here should be removed.
Author response: It has been revised as suggested and revisions are highlighted by yellow background color.
- Reference: Remove the old citation and replace the new one. Older than 2010 should be removed.
Author response: It has been revised as suggested.
- -Check format according to journal guidelines.
Author response: It has been revised as suggested.

Round 2
Reviewer 1 Report
Thank you for the answers and the revisions of the manuscript.
I only have one minor concern left:
Figure 1. legend Relative hydrolytic activity needs some explanation somewhere in the text.
Reviewer 3 Report
Dear Authors,
The overall revision version is complete, and I recommend that it be published as is.
Regards,